# The COVID-19 pandemic and health-related quality of life across 13 high- and low-middle-income countries: A cross-sectional analysis

Mara Violato [1,2] *, Jack Pollard [1], Andrew Lloyd [3,4], Laurence S. J. Roope [1,2], Raymond Duch [5], Matias Fuentes Becerra [6], Philip M. Clarke [1,2,7]

1 Health Economics Research Centre, Nuffield Department of Population Health, University of Oxford, Oxford, United Kingdom, 2 National Institute for Health Research Oxford Biomedical Research Centre, John Radcliffe Hospital, Oxford, United Kingdom, 3 Acaster Lloyd Consulting Ltd, London, England, United Kingdom, 4 Global Health Economics Centre, London School of Hygiene & Tropical Medicine, London, United Kingdom, 5 Nuffield College, University of Oxford, Oxford, United Kingdom, 6 Department of Economics, School of Economics and Business, University of Chile, Santiago, Región Metropolitana, Chile, 7 Centre For Health Policy, University of Melbourne, Parkville, Victoria, Australia

* mara.violato@dph.ox.ac.uk

**Data Availability Statement:** All 42 files are available from the Dataverse Repository database (accession number(s) https://doi.org/10.7910/

## Abstract

### Background

Most research on the Coronavirus Disease 2019 (COVID-19) health burden has focused on confirmed cases and deaths, rather than consequences for the general population's health-related quality of life (HRQoL). It is also important to consider HRQoL to better understand the potential multifaceted implications of the COVID-19 pandemic in various international contexts. This study aimed to assess the association between the COVID-19 pandemic and changes in HRQoL in 13 diverse countries.

### Methods and findings

Adults (18+ years) were surveyed online (24 November to 17 December 2020) in 13 countries spanning 6 continents. Our cross-sectional study used descriptive and regression-based analyses (age adjusted and stratified by gender) to assess the association between the pandemic and changes in the general population's HRQoL, measured by the EQ-5D-5L instrument and its domains (mobility, self-care, usual activities, pain/discomfort, and anxiety/depression), and how overall health deterioration was associated with individual-level (socioeconomic, clinical, and experiences of COVID-19) and national-level (pandemic severity, government responsiveness, and effectiveness) factors. We also produced country-level quality-adjusted life years (QALYs) associated to COVID-19 pandemic-related morbidity. We found that overall health deteriorated, on average across countries, for more than one-third of the 15,480 participants, mostly in the anxiety/depression health domain, especially for younger people (<35 years old) and females/other gender. This translated overall into a 0.066 mean "loss" (95% CI: −0.075, −0.057; p-value < 0.001) in the EQ-5D-5L index, representing a reduction of 8% in overall HRQoL. QALYs lost associated with morbidity were 5 to 11 times greater than QALYs lost based on

DVN/PMV0TG) https://dataverse.harvard.edu/dataset.xhtml?persistentId=doi:10.7910/DVN/PMV0TG.

**Funding:** PC received funding from the NIHR Biomedical Research Centre at the University of Oxford, award number NIHR-BRC-1215-20008. MV received funding from the EuroQol Research Foundation, award number 118-2020RA, and the NIHR Applied Research Collaboration Oxford and Thames Valley, award number NIHR200172. The views expressed are those of the authors and not necessarily those of the NIHR or the Department of Health and Social Care or the EuroQol Research Foundation. The funders had no role in study design, data collection and analysis, decision to publish, or preparation of the manuscript.

**Competing interests:** I have read the journal's policy and the authors of this manuscript have the following competing interests: AL is a Board member of the EuroQol Group which is responsible for licensing the EQ-5D. The EQ-5D was used in the present study. The other authors have declared that no competing interests exist.

**Abbreviations:** CANDOUR, Covid-19 vAccine preference anD Opinion sURvey; CHI, Containment and Health Index; COVID-19, Coronavirus Disease 2019; ESI, Economic Support Index; GE, Government Effectiveness; GRI, Government Response Index; HRQoL, health-related quality of life; ICG, income classification group; OR, odds ratio; OxCGRT, Oxford COVID-19 Government Response Tracker; PCHC, Paretian Classification of Health Change; QALY, quality-adjusted life year; SI, Stringency Index; WGI, Worldwide Governance Indicators.

COVID-19 premature mortality. A limitation of the study is that participants were asked to complete the prepandemic health questionnaire retrospectively, meaning responses may be subject to recall bias.

## Conclusions

In this study, we observed that the COVID-19 pandemic was associated with a reduction in perceived HRQoL globally, especially with respect to the anxiety/depression health domain and among younger people. The COVID-19 health burden would therefore be substantially underestimated if based only on mortality. HRQoL measures are important to fully capture morbidity from the pandemic in the general population.

---

### Author summary

#### Why was this study done?

- The health burden of the Coronavirus Disease 2019 (COVID-19) pandemic has been closely tracked in terms of confirmed cases and deaths due to the virus, but data on the relationship between the COVID-19 pandemic and health-related quality of life (HRQoL) of the general population globally is still quite limited.

- Existing studies have mainly focused on single countries and used relatively small convenience samples, making cross-country comparison difficult.

- It is important to understand similarities and differences in the association between the COVID-19 pandemic and HRQoL in different countries around the world, because this information can help policy-makers to find the most appropriate interventions that work in diverse settings, health conditions, and populations, to combat the ongoing COVID-19 pandemic and potential future pandemics.

#### What did the researchers do and find?

- We asked 15,536 participants from a diverse group of 13 countries (approx. 1,200 per country) to answer an online survey in which we asked questions about sociodemographic characteristics, health, and perceived HRQoL prior to and during the pandemic.

- We found that the COVID-19 pandemic was associated with significantly worse HRQoL for more than one-third of respondents, with anxiety/depression being the aspect of health that worsened the most, especially for younger people (<35 years) and females.

- Overall, our results suggest that there was an 8% reduction in perceived HRQoL, and the health burden, measured in quality-adjusted life years, associated with morbidity was considerably higher than that due to premature, COVID-19-related, mortality.

**What do these findings mean?**

- The health burden of the COVID-19 pandemic may be substantially underestimated if we consider mortality alone.

- HRQoL measures are important to fully capture the considerable health burden of the COVID-19 pandemic and its associated containment policies.

- Our results provide benchmark evidence for countries at different stages in the pandemic and can help inform public health measures and economic policies in the event of other health shocks in the future.

## Introduction

The impact of the ongoing Coronavirus Disease 2019 (COVID-19) pandemic has proved to be wide and pervasive. From a population health perspective, the impact of COVID-19 has been closely tracked in terms of confirmed cases and deaths [1,2]. This, however, underestimates the population health burden because it cannot capture the wider, longer-term, multifaceted association that the COVID-19 pandemic and related containment measures have had with the general population's health-related quality of life (HRQoL).

Collecting and analysing self-reported measures of health during the COVID-19 pandemic should be a priority for the research community [3], to better understand how multiple factors are associated not only with those directly infected with COVID-19, but also with the wider population. Different regions/countries, population groups, and cultures experience different levels of exposure to COVID-19; different availability of healthcare, economic resources, and governmental containment policies; and different socioeconomic impacts. Documenting differences and/or similarities in HRQoL across countries as the pandemic unfolds and the COVID-19 vaccine/booster rollout proceeds is, therefore, important to inform policy-makers about the most appropriate interventions in multiple settings, health conditions, and populations, and assess their impact.

In this spirit, a growing number of studies have emerged [4–7], which report how HRQoL has changed during the pandemic compared with prepandemic levels using the EQ-5D-5L, a well-validated, preference-based, generic health instrument [8]. Use of the EQ-5D-5L across countries provides a standardized approach to measuring health within and across nations. The EQ-5D-5L index, which varies between 0 (dead) and 1 (full health), can be used to estimate quality-adjusted life years (QALYs), a measure of health burden widely used in economic evaluations in healthcare. In addition, the EQ-5D-5L is also used commonly in studies of population health.

Recent studies using the EQ-5D-5L to quantify the impact of COVID-19 have reported a deterioration of HRQoL, in general and, more specifically, in the anxiety/depression domain [4–7,9]. However, they mainly focused on high-income countries [5–7] with fewer investigations referring to middle-income countries [4,10,11] and—to our knowledge—none in low-income countries. Previous studies were limited by small sample sizes, use of convenience sampling methods, restricted clinical populations, or did not include multiple cross-country comparisons.

QALYs derived from standard life table methods, adjusting for comorbidities and EQ-5D-derived HRQoL, have been recently used to assess the value of a COVID-19-related death. A recent study [12] suggested a COVID-19-related death was equal to around 5 QALYs lost, on

average, across the United Kingdom, United States, Canada, Norway, and Israel. A QALY loss ranging between 3.2 and 6.5 QALYs, depending on assumptions on standardized mortality ratio and impact of comorbidities on HRQoL, was instead indicated in a Spanish study [13], while a Dutch study estimated that QALYs lost because of COVID-19 mortality are on average 3.9 per death for men and 3.5 for women [14].

In this context, the primary aim of our study was to describe and assess the association of the COVID-19 pandemic with the change in HRQoL of the general population in 13 countries (Australia, Brazil, Canada, Chile, China, Colombia, France, India, Italy, Spain, Uganda, USA, and UK) participating, at the end of 2020, in the Covid-19 vAccine preference anD Opinion sURvey (CANDOUR) study [15]. Our prespecified hypotheses of health deterioration are detailed in the study protocol [16]. Our secondary aim was to use observed changes in the EQ-5D-5L index to estimate QALYs lost associated with morbidity at population level by country.

## Methods

### Study design, setting, and population

This cross-sectional investigation was embedded within the first wave of the CANDOUR study [15], a longitudinal, web-based, multicountry survey. Anonymous online surveys were completed by adults aged 18 years or more across the 13 participating countries, which are very diverse in their social and economic settings, between 24 November 2020 and 17 December 2020, i.e., within the first year of the COVID-19 pandemic. Except for India and Uganda, where samples were not nationally representative, including mainly urban settings, quota sampling was adopted to obtain representative samples in terms of age, education, gender, and geography in each country. For countries where imbalances persisted, a poststratification weighting was implemented. Full technical details of the CANDOUR study sample, including sampling and weighting procedures, are described at length in one of our cognate publications to which we point the interested reader [17]. This study protocol was preregistered [16] and approved by the University of Oxford Medical Sciences Interdivisional Research Ethics Committee (ID: R72328/RE001). All participants provided informed written consent at the beginning of the survey. This study is reported as per the Strengthening the Reporting of Observational Studies in Epidemiology (STROBE) guideline (S1 Table).

### External data sources

We used data from the Oxford COVID-19 Government Response Tracker (OxCGRT) database [1,18] and "Worldwide Governance Indicators (WGI)" [19,20], linked to CANDOUR data, to explore the relationship between national policies/government effectiveness and perceived health of study participants. Population estimates by country and age categories were obtained from the Global Burden of Disease Study 2019 [21].

### Procedures

Participants' health was captured by the EQ-5D-5L [8], which covers 5 domains: mobility (i.e., walking), self-care (i.e., washing or dressing), usual activities (i.e., work, study, housework, and leisure activities), pain/discomfort, and anxiety/depression. Each domain has 5 ordered levels, from no (1) to extreme (5) problems. Participants rated their current health at the time of the survey, as per standard EQ-5D instrument, but also retrospectively (under EuroQol Agreement 159150), thinking to the pre-COVID-19 pandemic period.

We compared the participants' EQ-5D-5L profiles at the time of the survey and pre-COVID-19 pandemic using the Paretian Classification of Health Change (PCHC) approach

[22]. The latter defines an EQ-5D-5L health profile as improved (worsened) with respect to another, if it improved (worsened) on at least one dimension and had not worsened (improved) on any other dimension. Each respondent's perceived change in health can be classified into 5 mutually exclusive categories: improved; worsened; "mixed" (i.e., health improved in at least one dimension, but worsened in at least one other); unchanged and equal to full health (i.e., all dimensions remained at level 1, indicating no problems); unchanged but different from full health (i.e., at least one dimension had a level higher than 1). EQ-5D-5L indices, measuring HRQoL, were generated using the UK [23] value set in the main analysis to enable comparability between countries, and the US [24] and India [25] value sets in the sensitivity analyses, incorporating in this way value sets from both high-income and middle-income countries. QALYs lost at population levels were estimated from the UK-valued EQ-5D-5L and external data on population sizes.

All CANDOUR variables included in the analyses are summarised in Table 1. Indicators of COVID-19 government responses at the national level and government performance included 4 composite indices from the OxCGRT database: overall Government Response Index (GRI), Containment and Health Index (CHI), Economic Support Index (ESI), Stringency Index (SI); and the Government Effectiveness (GE) indicator from the WGI database. Indices definitions and further details can be found in the Supporting information below (S3 Table). Pandemic severity was proxied by quintiles of incident cases and deaths (7-day average prior to the survey date).

## Statistical analyses

Descriptive analyses were conducted. Continuous variables were reported as mean values and standard deviations and, in the case of differences between variables, 95% confidence intervals. Categorical variables were presented as counts and 95% confidence intervals. Statistical comparisons were made using $t$ tests to compare mean differences, and equality of proportions tests to compare differences in proportions.

Results from the PCHC approach applied to the EQ-5D-5L profile data were reported descriptively, stratified by continent and World Bank income classification group (ICG). For the category "health worsened," descriptive results were reported by EQ-5D-5L dimension, ICG, and individual-level comorbidities. Univariable and multivariable logistic regressions were conducted to explore the association between perceived worsened health (1: health worsened; 0: otherwise), and 3 sets of potential predictors. These included individual-level socioeconomic and clinical factors; individual-level experiences of/exposure to COVID-19; and macro-level variables, i.e., national-level government responsiveness to COVID-19, pandemic severity, and government effectiveness. Because of small numbers, the respondents who reported their gender as "Other" were arbitrarily combined with those who identified as "Female" for the purposes of the analysis. Exploration of data suggested that there was not a clear pattern and, in fact, sample sizes were too small to draw any meaningful conclusion. The category "Other" included all individuals who did not identified either as "Male" or "Female." For each set of predictors, we first estimated the unadjusted association with each predictor (Model 1); we then adjusted it by age and country (Model 2); and, finally, predictors whose association had a $p$-value of <0.10 were included in the multivariable logistic regression (Model 3). Model 3 was not performed for national-level variables, as some indicators were nested within each other. Logistic regression results were reported as odds ratios (ORs) with 95% confidence intervals and stratified by gender. A $p$-value of <0.05 was considered statistically significant. All statistical analyses were conducted using Stata 17.0 (StataCorp LP; College Station, TX).

**Table 1. Sociodemographic and clinical characteristics, overall and by continent.**

| | Overall sample | Africa | Asia | Europe | North America | Oceania | South America |
|---|---|---|---|---|---|---|---|
| No. sampled–N | 15,480 | 1,038 | 2,481 | 4,537 | 2,294 | 1,358 | 3,772 |
| Gender—% (95% CI) | | | | | | | |
| Male | 51.49 (50.40–52.58) | 73.41 (70.64–76.01) | 60.99 (57.99–63.91) | 48.19 (46.67–49.71) | 51.28 (49.07–53.49) | 46.01 (43.18–48.87) | 45.27 (42.30–48.28) |
| Female and Other[†] | 48.51 (47.42–49.60) | 26.59 (23.99–29.36) | 39.01 (36.09–42.01) | 51.81 (50.29–53.33) | 48.72 (46.51–50.93) | 53.99 (51.13–56.82) | 54.73 (51.72–57.70) |
| Missing | 0 (0.00) | 0 (0.00) | 0 (0.00) | 0 (0.00) | 0 (0.00) | 0 (0.00) | 0 (0.00) |
| Age (years)—Mean (SD) | 44.14 (16.49) | 29.11 (7.02) | 42.00 (16.04) | 48.16 (15.94) | 46.68 (17.11) | 46.01 (17.47) | 42.63 (15.72) |
| Education—% (95% CI) | | | | | | | |
| Primary or less | 23.56 (22.27–24.90) | 3.85 (2.84–5.21) | 52.10 (49.04–55.16) | 9.46 (8.54–10.45) | 4.48 (3.67–5.46) | 25.67 (22.96–28.59) | 38.00 (34.59–41.54) |
| Secondary | 41.27 (40.26–42.29) | 31.89 (29.12–34.79) | 17.99 (16.29–19.82) | 51.41 (49.89–52.93) | 55.58 (53.42–57.72) | 46.35 (43.51–49.22) | 36.45 (34.06–38.92) |
| University | 33.22 (32.35–34.10) | 62.91 (59.93–65.80) | 29.04 (26.86–31.32) | 37.57 (36.12–39.05) | 38.90 (36.85–40.98) | 26.60 (24.52–28.78) | 21.49 (20.02–23.04) |
| Missing | 1.95 (1.75–2.18) | 1.35 (0.80–2.26) | 0.87 (0.57–1.32) | 1.56 (1.25–1.96) | 1.04 (0.70–1.54) | 1.37 (0.88–2.14) | 4.05 (3.46–4.75) |
| Employment[*] - % (95% CI) | | | | | | | |
| Employed | 47.08 (46.01–48.16) | 43.06 (40.08–46.10) | 57.51 (54.19–60.77) | 42.59 (41.10–44.09) | 53.02 (50.80–55.23) | 45.62 (42.81–48.46) | 43.66 (40.88–46.48) |
| Unemployed | 9.71 (9.17–10.27) | 26.97 (24.36–29.76) | 3.58 (2.74–4.68) | 7.81 (7.05–8.64) | 7.50 (6.32–8.89) | 8.29 (6.80–10.07) | 13.11 (11.69–14.68) |
| Pension/capital income | 11.87 (10.97–12.84) | 0.19 (0.05–0.77) | 18.75 (15.99–21.86) | 12.20 (11.25–13.22) | 19.80 (18.10–21.61) | 0 (0.00) | 9.61 (7.09–12.89) |
| Other | 19.60 (18.63–20.61) | 5.49 (4.26–7.05) | 19.55 (16.76–22.69) | 10.00 (9.08–11.00) | 17.94 (16.23–19.78) | 40.77 (37.96–43.64) | 28.43 (25.69–31.35) |
| Missing | 11.74 (11.13–12.39) | 24.28 (21.76–26.98) | 0.60 (0.36–1.00) | 27.40 (26.04–28.81) | 1.74 (1.25–2.43) | 5.32 (4.11–6.87) | 5.18 (3.89–6.88) |
| Loss of income due to COVID-19 - % (95% CI) | | | | | | | |
| Yes | 43.07 (41.97–44.18) | 88.73 (86.66–90.51) | 42.43 (39.42–45.49) | 28.45 (27.11–29.83) | 31.29 (29.31–33.36) | 27.15 (24.76–29.68) | 61.40 (58.66–64.08) |
| No | 52.28 (51.18–53.38) | 8.96 (7.37–10.86) | 54.75 (51.66–57.81) | 66.32 (64.88–67.73) | 64.25 (62.12–66.33) | 68.49 (65.85–71.02) | 32.57 (30.18–35.07) |
| Don't know | 2.36 (1.97–2.83) | 0.48 (0.20–1.15) | 1.58 (1.09–2.29) | 2.96 (2.49–3.52) | 2.62 (1.94–3.53) | 1.81 (1.22–2.69) | 2.71 (1.55–4.68) |
| Missing | 2.29 (2.04–2.57) | 1.83 (1.17–2.85) | 1.24 (0.87–1.75) | 2.27 (1.86–2.78) | 1.83 (1.34–2.50) | 2.55 (1.77–3.64) | 3.31 (2.65–4.13) |
| Believed to have had COVID-19 - % (95% CI) | | | | | | | |
| Yes | 15.98 (15.31–16.68) | 18.79 (16.52–21.28) | 23.67 (21.58–25.89) | 12.75 (11.76–13.82) | 12.07 (10.71–13.57) | 9.93 (8.51–11.55) | 18.60 (16.89–20.45) |
| No | 69.42 (68.49–70.34) | 62.14 (59.15–65.04) | 72.80 (70.44–75.05) | 70.01 (68.60–71.38) | 76.26 (74.35–78.08) | 84.44 (82.44–86.26) | 58.93 (56.17–61.63) |
| Don't know | 10.75 (10.12–11.42) | 19.08 (16.80–21.58) | 0.43 (0.15–1.18) | 10.29 (9.42–11.23) | 6.19 (5.27–7.26) | 0 (0.00) | 22.47 (20.36–24.73) |
| Missing | 3.84 (3.53–4.18) | 0 (0.00) | 3.10 (2.47–3.88) | 6.95 (6.22–7.76) | 5.48 (4.51–6.64) | 5.63 (4.49–7.03) | 0 (0.00) |
| Tested positive for COVID-19 - % (95% CI) | | | | | | | |
| Yes | 10.53 (9.89–11.22) | 6.45 (5.11–8.12) | 20.44 (18.25–22.83) | 7.41 (6.66–8.24) | 7.00 (6.00–8.15) | 8.54 (7.20–10.11) | 11.76 (10.04–13.72) |

*(Continued)*

**Table 1.** (Continued)

| | Overall sample | Africa | Asia | Europe | North America | Oceania | South America |
|---|---|---|---|---|---|---|---|
| **No. sampled–N** | **15,480** | **1,038** | **2,481** | **4,537** | **2,294** | **1,358** | **3,772** |
| **Gender—% (95% CI)** | | | | | | | |
| No | 87.65 (86.94–88.33) | 92.68 (90.93–94.11) | 78.01 (75.57–80.27) | 90.63 (89.71–91.47) | 90.66 (89.32–91.85) | 89.13 (87.37–90.67) | 86.67 (84.67–88.46) |
| Don't know | 0.87 (0.71–1.06) | 0.87 (0.45–1.66) | 0.51 (0.21–1.23) | 0.85 (0.62–1.17) | 0.65 (0.39–1.08) | 0 (0.00) | 1.57 (1.14–2.15) |
| Missing | 0.95 (0.79–1.13) | 0 (0.00) | 1.05 (0.71–1.54) | 1.11 (0.82–1.49) | 1.69 (1.15–2.47) | 2.33 (1.62–3.33) | 0 (0.00) |
| **Relative infected with COVID-19 - % (95% CI)** | | | | | | | |
| Yes | 30.07 (29.01–31.14) | 28.52 (25.85–31.34) | 31.27 (28.85–33.79) | 26.56 (25.23–27.92) | 22.07 (20.32–23.92) | 11.00 (9.48–12.72) | 45.66 (42.65–48.70) |
| No | 64.92 (63.83–65.99) | 61.95 (58.95–64.85) | 66.08 (63.45–68.61) | 68.62 (67.19–70.02) | 72.57 (70.57–74.48) | 85.62 (83.69–87.36) | 48.40 (45.47–51.35) |
| Don't know | 3.37 (3.02–3.76) | 9.54 (7.89–11.48) | 1.00 (0.57–1.75) | 2.64 (2.20–3.16) | 2.35 (1.81–3.06) | 0 (0.00) | 5.94 (4.84–7.26) |
| Missing | 1.65 (1.44–1.88) | 0 (0.00) | 1.65 (1.21–2.24) | 2.18 (1.77–2.69) | 3.01 (2.26–3.99) | 3.39 (2.56–4.47) | 0 (0.00) |
| **Friend/colleague infected with COVID-19 - % (95% CI)** | | | | | | | |
| Yes | 42.37 (41.33–43.42) | 50.48 (47.44–53.52) | 32.67 (30.21–35.23) | 46.05 (44.53–47.57) | 31.09 (29.08–33.17) | 13.18 (11.52–15.04) | 59.47 (56.30–62.56) |
| No | 51.75 (50.67–52.83) | 41.23 (38.27–44.26) | 64.67 (62.00–67.25) | 47.61 (46.09–49.13) | 62.28 (60.10–64.40) | 82.88 (80.81–84.78) | 33.50 (30.33–36.83) |
| Don't know | 3.93 (3.59–4.32) | 8.29 (6.76–10.12) | 1.01 (0.54–1.89) | 3.52 (3.02–4.11) | 3.18 (2.54–3.99) | 0 (0.00) | 7.03 (6.00–8.23) |
| Missing | 1.95 (1.72–2.20) | 0 (0.00) | 1.65 (1.21–2.24) | 2.82 (2.34–3.39) | 3.45 (2.64–4.50) | 3.94 (3.03–5.10) | 0 (0.00) |
| **Know of someone dead from COVID-19 - % (95% CI)** | | | | | | | |
| Yes | 37.02 (35.94–38.10) | 69.85 (66.98–72.56) | 32.37 (29.90–34.94) | 31.34 (29.95–32.76) | 21.20 (19.51–23.00) | 12.66 (11.01–14.51) | 56.25 (53.35–59.11) |
| No | 59.77 (58.68–60.85) | 27.55 (24.92–30.35) | 64.83 (62.16–67.41) | 64.92 (63.46–66.35) | 74.72 (72.78–76.56) | 84.63 (82.62–86.44) | 41.08 (38.26–43.95) |
| Don't know | 1.58 (1.37–1.81) | 2.60 (1.79–3.77) | 0.55 (0.32–0.94) | 1.70 (1.34–2.14) | 1.13 (0.77–1.66) | 0 (0.00) | 2.67 (2.11–3.38) |
| Missing | 1.64 (1.43–1.87) | 0 (0.00) | 2.26 (1.73–2.93) | 2.04 (1.65–2.54) | 2.95 (2.20–3.93) | 2.71 (1.98–3.71) | 0 (0.00) |
| **Comorbidities—% (95% CI)** | | | | | | | |
| Diabetes | 11.73 (10.84–12.67) | 1.82 (1.15–2.86) | 17.51 (15.63–19.57) | 8.69 (7.86–9.60) | 14.69 (13.19–16.33) | 13.98 (12.10–16.09) | 11.65 (8.85–15.19) |
| Hypertension | 18.94 (17.97–19.95) | 4.34 (3.23–5.80) | 18.76 (16.47–21.30) | 17.62 (16.47–18.83) | 25.20 (23.28–27.21) | 23.72 (21.31–26.31) | 19.10 (16.21–22.38) |
| Heart disease | 4.50 (4.11–4.92) | 1.61 (0.99–2.62) | 7.88 (6.58–9.42) | 3.99 (3.43–4.65) | 5.03 (4.10–6.16) | 5.48 (4.28–6.99) | 2.98 (2.26–3.92) |
| Asthma | 8.44 (7.90–9.00) | 4.44 (3.32–5.91) | 6.63 (5.21–8.40) | 8.32 (7.51–9.22) | 12.54 (11.08–14.15) | 15.41 (13.33–17.75) | 5.84 (4.84–7.03) |
| Allergies[+] | 18.17 (17.39–18.98) | 26.64 (23.98–29.48) | 13.67 (11.48–16.20) | 14.52 (13.46–15.64) | 25.88 (23.96–27.90) | 18.68 (16.54–21.02) | 18.43 (16.56–20.45) |
| Kidney disease | 2.16 (1.77–2.63) | 1.11 (0.62–1.99) | 4.08 (3.05–5.45) | 1.22 (0.92–1.61) | 2.15 (1.57–2.93) | 1.65 (1.05–2.57) | 2.50 (1.42–4.38) |
| Other condition | 9.01 (8.37–9.69) | 4.34 (3.23–5.80) | 3.33 (2.55–4.35) | 9.97 (9.08–10.93) | 13.01 (11.51–14.68) | 16.02 (13.89–18.40) | 7.93 (6.15–10.17) |

(*Continued*)

**Table 1.** (*Continued*)

| | Overall sample | Africa | Asia | Europe | North America | Oceania | South America |
|---|---|---|---|---|---|---|---|
| **No. sampled–N** | **15,480** | **1,038** | **2,481** | **4,537** | **2,294** | **1,358** | **3,772** |
| **Gender—% (95% CI)** | | | | | | | |
| No comorbidity | 49.55 (48.45–50.66) | 62.56 (59.50–65.52) | 53.53 (50.32–56.71) | 52.28 (50.73–53.82) | 42.15 (39.95–44.39) | 43.85 (41.01–46.73) | 46.61 (43.70–49.54) |

*N* = actual sample size; % = weighted percentage; Mean = weighted mean; SD = weighted standard deviation.

*Employment variable missing for all French respondents.

†Because of small numbers, the respondents who reported their gender as "Other" were arbitrarily combined with those who identified as "Female" for the purposes of the analysis.

⁺The category "Allergies" refers to any kind of allergy, such as food allergy (e.g., lactose, eggs), plants and flowers (e.g., daisy), hay-fever, etc.

Descriptive statistics by EQ-5D-5L dimension that worsened were also reported by age and gender, with anxiety/depression (the most prevalent) also presented by continent. Mean EQ-5D-5L indices prepandemic, and at the time of the survey, and their mean difference were reported by country alongside their 95% confidence intervals.

Finally, we estimated QALYs lost associated with morbidity at the population level by country. We first calculated the HRQoL change, using a single value set (UK value set), to enable comparability across countries, from pre- to during pandemic by age and country. Under the simplifying assumption that changes remained stable for a year, we then multiplied them by the total national population in each age group. The resulting country-level QALY loss associated to COVID-19 pandemic-related morbidity was then expressed as a ratio of country-level QALY loss due to COVID-19 premature mortality. Country-level QALY loss due to COVID-19-related death was estimated by drawing on the recent literature expressing the value of a COVID-19 death in terms of QALYs lost [12–14]. In particular, we used the highest and lowest estimates of QALYs lost per death [13], so to determine the widest possible range of values, and multiplied it by each of the confirmed country-level COVID-19 deaths a year into the pandemic [1,18].

## Results

### Characteristics of study participants

On average, 1,195 individuals per country participated in the CANDOUR study, a total of 15,536 participants. Of these, all completed the EQ-5D-5L descriptive system. We restricted analyses to respondents with complete data on age and gender, which marginally reduced the sample size to 15,480 individuals (S1 Fig). Missing data on other categorical variables were included as an additional category (Table 1).

The profile of the overall sample is shown in Table 1. Across countries, on average, 43% of respondents incurred loss of income due to COVID-19. On average, less than 20% of respondents had (or believed they had) been infected with COVID-19, while 30% to 42% experienced it through family/friends. There were differences at continent and country levels (Tables 1 and S2). The average level of government responsiveness (GRI, CHI, ESI, and SI indicators) ranged from 65 to 69 (scale 0 to 100), with Uganda scoring the lowest and Chile and Italy scoring the highest. Severity of pandemic varied depending on the country population and the stage of the pandemic, with Australia and China reporting the lowest mean number of incident cases and deaths (7-day means), respectively, and the US the highest (S3 Table).

### Paretian classification of health change

Although, on average, health remained unchanged for about 48% of respondents (Fig 1 and S4 and S5 Tables), it worsened for more than one-third (35%). At the continent level, only Asia (32%) and Oceania (25%) had a lower percentage. Worsened health remained prevalent when stratifying results by ICG (S2 Fig).

Among those who perceived their health as worsened ($n$ = 5,632), the highest and lowest changes in EQ-5D-5L domains were in anxiety/depression (81%) and self-care (16%), respectively (S3 Fig). Reporting worsened health increased with the number of long-term health conditions (S4 Fig), and was greatest in lower country income groups (S5 Fig), except for the anxiety/depression domain.

### Factors associated with PCHC category "health worsened"

The associations between worsened health and participants' socioeconomic and clinical characteristics, adjusted by age and country, were similar across genders (S6 Fig and S6 Table—Models 2/3). For female/other gender, the odds of worsened health were significantly increased by 51% and 200% for those who completed secondary and tertiary education, respectively, compared with those less educated, while for males, a lower increase of 30% was found for university graduates. No significant association with employment status was found among those who responded. Income losses due to the pandemic and having long-term conditions were associated, respectively, with significantly higher odds of worsened health for both males (around 60% and 70%) and females/other (about 50% and 66%).

Across genders, the odds of worsened health significantly increased by about 30% for almost all individual-level experiences of/exposure to COVID-19 (S7 Fig and S7 Table—Models 2/3). There were no significant associations between worsened health and indicators of government responsiveness to, and severity of, COVID-19 at national levels (S8 Fig and S8 Table—Model 2)

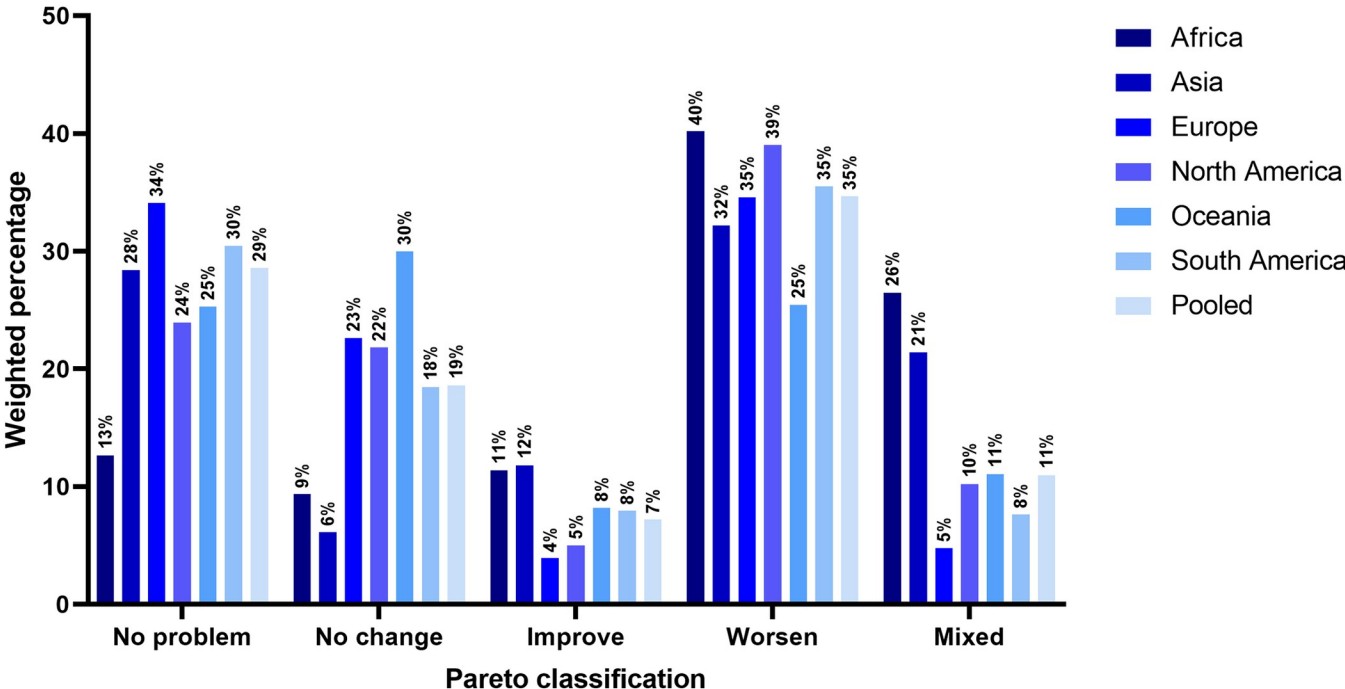

**Fig 1. Paretian Classification of Health Change (PCHC), overall and by continent.**

for female/other gender. Statistically significant associations existed among male respondents, with odds of worsened health significantly decreasing for higher values of the GRI and ESI indices distribution, and indicators of pandemic severity, but significantly increasing with higher values of the SI index. Male respondents living in countries with higher levels of government effectiveness (S9 Fig and S9 Table—Model 2) were significantly less likely to report worsened health, but the opposite was true for females/others. Results on these macro-indicators reflected the wider context in which each country was at the time of the survey.

### Deterioration of health by EQ-5D domains

When looking at individual EQ-5D domains, the highest and the lowest observed change were in anxiety/depression (33.8%; $n = 5,525$) and self-care (10%; $n = 1,553$). For anxiety/depression (Fig 2), deterioration was more commonly reported by younger generations, with males and females/others aged 18 to 24 reporting worsened health 13 and 14 percentage points, respectively, higher than those aged 65+. However, perceived deterioration of mental health was, on average, 4 percentage points higher for females/others than males throughout the age distribution. Similar results held across continents, with Africa (Uganda) and South America (Brazil, Chile, Colombia) being outliers (S10 Fig). For all other health domains, worsening was reported in greater percentage by younger groups, but the percentage of males who reported deterioration in health was, on average, greater than the corresponding percentage reported by females (S11 Fig).

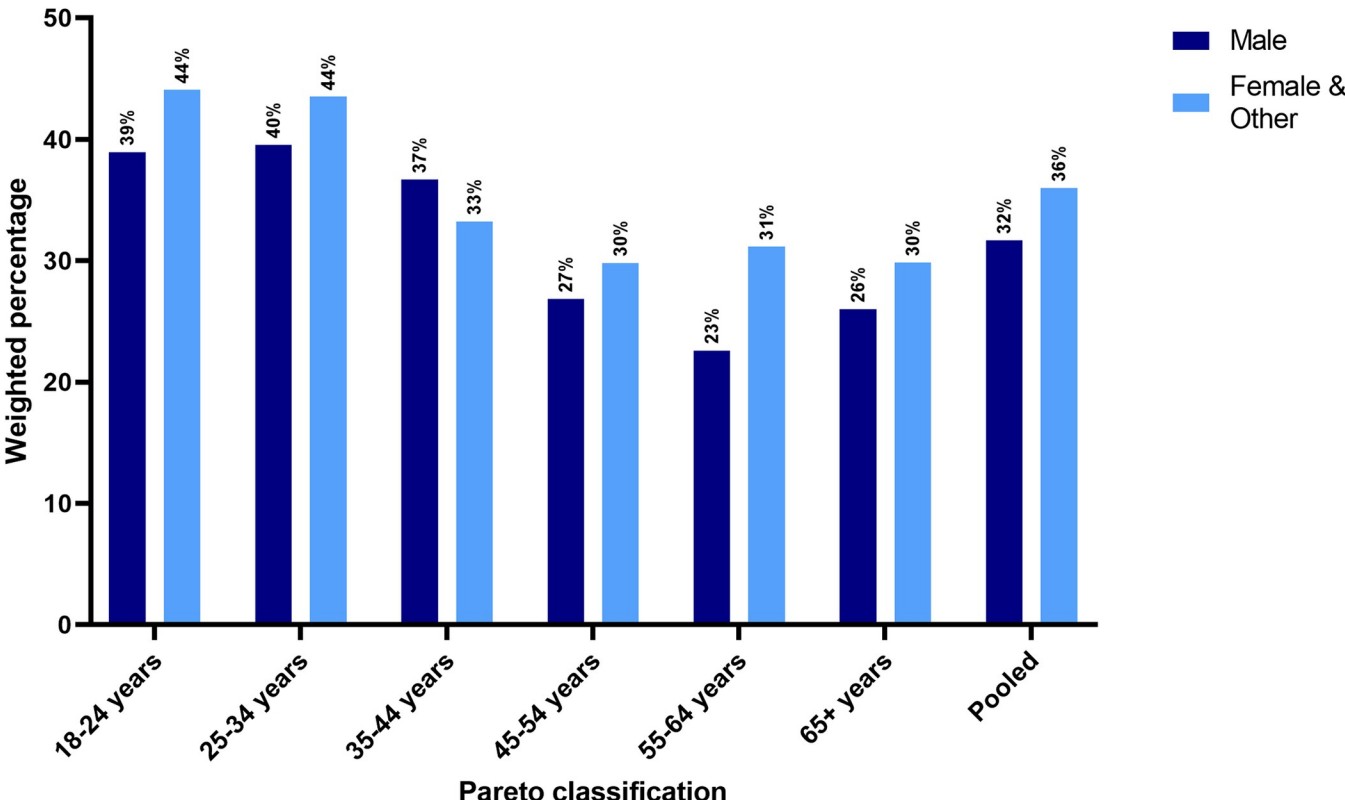

**Fig 2. Percentage of each age group reporting "Worsened" anxiety/depression, by gender.** Note: Because of small numbers, the respondents who reported their gender as "Other" were arbitrarily combined with those who identified as "Female" for the purposes of the analysis.

### HRQoL changes

Responses reporting Level 1 "no problems" decreased significantly during the pandemic across all health domains (S12 Fig and S10 Table), with the highest decrease in anxiety/depression (mean difference; −17%; 95% CI: −18%, −16%; p-value: <0.001). Conversely, there were significantly more responses reporting Levels 2 to 5 (i.e., slight to extreme problems) during than before the pandemic (S10 Table).

Placing UK valuations [23] on the EQ-5D-5L health profiles (Table 2), we found that mean HRQoL significantly deteriorated by 0.066 (95% CI: −0.075, −0.057; p-value: <0.001) during the pandemic. Only China showed nonstatistically significant changes. Overall, significant decrements pre-during pandemic in mean HRQoL were slightly lower for males (0.063) than for females/others (0.069) (S11 and S12 Tables). There were no significant HRQoL changes for males in China, Colombia, and Chile or for females/others in China (S11 and S12 Tables). Results remained consistent when using the US value set (S13–S15 Tables) and India value set (S16–S18 Tables). Higher HRQoL decrements occurred among the younger age groups (S19 Table).

### QALY loss at the population level

After extrapolating mean differences in HRQoL to the population of each country, the median ratio across all countries except China (S19 Table), of "QALYs lost associated with morbidity/ QALYs lost due to COVID-19 mortality" (Table 3) was 5 and 11 using the highest and lowest estimates [13], respectively, of QALYs lost per death [12–14].

### Discussion

While deaths associated with the COVID-19 pandemic have been widely reported for most countries globally, there has been much less focus on how the pandemic and associated containment measures have affected other aspects of health. Generic quality of life measures, such as the EQ-5D, are now widely used to measure health, and this instrument has been shown to

**Table 2. Mean difference in EQ-5D-5L index (utility) pre-COVID-19 and at time of survey, UK value set.**

| Country | Utility pre-COVID-19 | | | Utility at survey | | | Utility difference | | |
|---|---|---|---|---|---|---|---|---|---|
| | N | Mean | SD | N | Mean | SD | Mean | 95% CI | p-Value |
| Australia | 1,358 | 0.772 | 0.261 | 1,358 | 0.718 | 0.293 | −0.053 | (−0.076, −0.030) | <0.001 |
| Brazil | 1,421 | 0.832 | 0.234 | 1,421 | 0.771 | 0.270 | −0.061 | (−0.085, −0.037) | <0.001 |
| Canada | 1,148 | 0.813 | 0.231 | 1,148 | 0.731 | 0.278 | −0.081 | (−0.102, −0.061) | <0.001 |
| Chile | 1,120 | 0.855 | 0.221 | 1,120 | 0.747 | 0.278 | −0.108 | (−0.186, −0.031) | 0.006 |
| China | 1,291 | 0.879 | 0.196 | 1,291 | 0.876 | 0.180 | −0.003 | (−0.032, 0.027) | 0.852 |
| Colombia | 1,231 | 0.859 | 0.245 | 1,231 | 0.830 | 0.239 | −0.030 | (−0.059, −0.000) | 0.048 |
| France | 1,142 | 0.845 | 0.233 | 1,142 | 0.800 | 0.249 | −0.046 | (−0.067, −0.024) | <0.001 |
| India | 1,190 | 0.708 | 0.353 | 1,190 | 0.600 | 0.358 | −0.108 | (−0.137, −0.080) | <0.001 |
| Italy | 1,080 | 0.858 | 0.198 | 1,080 | 0.808 | 0.239 | −0.051 | (−0.070, −0.031) | <0.001 |
| Spain | 1,152 | 0.902 | 0.175 | 1,152 | 0.851 | 0.192 | −0.050 | (−0.066, −0.035) | <0.001 |
| UK | 1,163 | 0.804 | 0.265 | 1,163 | 0.751 | 0.281 | −0.053 | (−0.076, −0.030) | <0.001 |
| US | 1,146 | 0.754 | 0.286 | 1,146 | 0.677 | 0.328 | −0.077 | (−0.107, −0.048) | <0.001 |
| Uganda | 1,038 | 0.730 | 0.357 | 1,038 | 0.570 | 0.405 | −0.160 | (−0.193, −0.127) | <0.001 |
| Overall | 15,480 | 0.817 | 0.261 | 15,480 | 0.751 | 0.294 | −0.066 | (−0.075, −0.057) | <0.001 |

N = actual sample size; Mean = weighted mean; SD = weighted standard deviation; CI = confidence interval.

**Table 3. QALYs loss due to premature mortality and QALYs associated with morbidity a year into the COVID-19 pandemic.**

| QALYs loss due to premature mortality and QALYs associated with morbidity a year into the COVID-19 pandemic–absolute values | | | | | |
|---|---|---|---|---|---|
| | | QALYs lost due to COVID-19 premature mortality | | QALYs lost associated with morbidity | Ratio of QALYs lost associated with morbidity and QALYs lost to death | |
| Country | Cumulative deaths up to 23/03/2021[*] | Using lower death value: 3.2 QALYs lost per death[+] | Using upper death value: 6.5 QALYs lost per death[+] | Using data CANDOUR study—QALYs | Ratio QALYs lost (morbidity/death)—death lower value | Ratio QALYs lost (morbidity/death)—death upper value |
| US | 543,452 | 1,739,046 | 3,532,438 | 18,861,848 | 11 | 5 |
| Brazil | 299,073 | 957,034 | 1,943,975 | 10,096,118 | 11 | 5 |
| India | 160,441 | 513,411 | 1,042,867 | 101,179,443 | 197 | 97 |
| UK | 126,370 | 404,384 | 821,405 | 2,863,851 | 7 | 3 |
| Italy | 105,879 | 338,813 | 688,214 | 2,439,259 | 7 | 4 |
| France | 92,921 | 297,347 | 603,987 | 2,421,390 | 8 | 4 |
| Spain | 73,744 | 235,981 | 479,336 | 1,893,951 | 8 | 4 |
| Colombia | 62,274 | 199,277 | 404,781 | 1,068,056 | 5 | 3 |
| Canada | 22,736 | 72,755 | 147,784 | 2,259,626 | 31 | 15 |
| Chile | 22,384 | 71,629 | 145,496 | 1,616,874 | 23 | 11 |
| Australia | 909 | 2,909 | 5,909 | 1,028,664 | 354 | 174 |
| Uganda | 334 | 1,069 | 2,171 | 3,027,621 | 2,833 | 1,395 |
| *Median* | *83,333* | *266,664* | *541,661* | *2,430,325* | *11* | *5* |
| QALYs loss due to premature mortality and QALYs associated with morbidity a year into the COVID-19 pandemic–values per million people | | | | | |
| | | QALYs lost due to COVID-19 premature mortality per million people | | QALYs lost associated with morbidity per million people | Ratio of QALYs lost associated with morbidity and QALYs lost to death | |
| Country | Cumulative deaths up to 23/03/2021[*] | Using lower death value: 3.2 QALYs lost per death[+] | Using upper death value: 6.5 QALYs lost per death[+] | Using data CANDOUR study—QALYs | Ratio QALYs lost (morbidity/death)—death lower value | Ratio QALYs lost (morbidity/death)—death upper value |
| US | 543,452 | 6,821 | 13,855 | 73,983 | 11 | 5 |
| Brazil | 299,073 | 6,012 | 12,212 | 63,423 | 11 | 5 |
| India | 160,441 | 550 | 1,117 | 108,363 | 197 | 97 |
| UK | 126,370 | 7,605 | 15,448 | 53,861 | 7 | 3 |
| Italy | 105,879 | 6,690 | 13,589 | 48,162 | 7 | 4 |
| France | 92,921 | 5,726 | 11,631 | 46,630 | 8 | 4 |
| Spain | 73,744 | 6,231 | 12,656 | 50,006 | 8 | 4 |
| Colombia | 62,274 | 5,796 | 11,773 | 31,063 | 5 | 3 |
| Canada | 22,736 | 2,491 | 5,060 | 77,360 | 31 | 15 |
| Chile | 22,384 | 5,174 | 10,509 | 116,785 | 23 | 11 |
| Australia | 909 | 152 | 309 | 53,780 | 354 | 174 |
| Uganda | 334 | 56 | 114 | 159,179 | 2,833 | 1,395 |
| *Median* | *83,333* | *5,761* | *11,702* | *58,642* | *11* | *5* |

[*]Source: Our world in data (https://ourworldindata.org/covid-deaths).
[+]Source: Hernando and colleagues [13].

be sensitive to COVD-19 in a number of countries [4–7]. In this study, we explored the association between the COVID-19 pandemic and HRQoL at the population level, across 13 countries. These countries jointly represent almost half of the adult world population, and diverse social and economic settings. We found that, 9 months into the pandemic, more than one-third of respondents perceived that their health had deteriorated since the prepandemic period. The greatest change was in anxiety/depression, especially for those <35 and females/others. The perceived deterioration translated overall into a 0.066 mean difference "loss" in the

EQ-5D-5L index, representing an 8% reduction in overall HRQoL. This deterioration is comparable to the impact of myocardial infarction or blindness in one eye in diabetic patients, which were estimated to reduce HRQoL by 0.055 and 0.074, respectively [26]. We also translated this into country-level QALYs lost, which can be compared with plausible estimates of QALYs lost from premature mortality [12–14]. A key result of our study is that for the median country in our sample (excluding China), the QALYs loss associated with morbidity is 5 (11) times greater than QALYs loss due to mortality, when using the highest (lowest) estimates of QALYs lost per COVID-19 death.

Caution needs to be exerted when interpreting our findings. Findings may be biased by the fact that participants were asked to complete the health questionnaire retrospectively, and, therefore, responses may be subject to "recall bias." Recall bias is a complex phenomenon whereby, depending on context, people may potentially overweight salient events in the past or underweight previous health problems with regard to their current health and demographic variables [27]. Current health may act as a lens or filter that affects how people recall the past [28]. The literature on recall accuracy using the EQ-5D is limited, and studies tend to be for specific populations and conditions [29–31]. Patients have also been found to recall their baseline HRQoL as being better than it actually was [32], which, in our study, would suggest the differences we observed within the first year of the COVID-19 pandemic were overestimated. Furthermore, retrospective EQ-5D has been used previously in surveys for both COVID-19 [33,34] and other diseases [35]. Compared with prepandemic norms in countries where EQ-5D-5L population norms are available, in Australia [36], Canada [37], China [38], Colombia [39], Spain [40], and the US [6], our retrospective HRQoL values are relatively low, even when the country-specific value sets were used (i.e., the US). If there is some downward bias in our baseline population average HRQoL estimates, all else equal, this may at least have the advantage of helping to mitigate any upward bias in our estimates of HRQoL fall. However, it also suggests that our sample may not be as representative as hoped, although HRQoL measures based on online responses have been found to be lower than face-to-face responses [6], and none of the aforementioned population norms are based on online responses. It is also worth noting that the EQ-5D-5L asks individuals to rate their current health status at the time of the survey [8], meaning responses are likely to be influenced by the context of the pandemic in each country at the time of the survey, conducted between 24 November 2020 and 17 December 2020. Given that this period coincided with a new wave of COVID-19 cases and deaths globally [41], respondents may have perceived their HRQoL as lower than they would have at other points during the pandemic, when new cases and deaths were lower.

Another note of caution is about the representativeness of our sample. While samples are generally representative on key sociodemographic/geographical factors in the included high-income countries, the same cannot be claimed for low/middle-income countries, with India and Uganda, for example, being primarily sampled from urban populations. This means that the included samples from these countries are not nationally representative, with the implication that the associated results are representative of a more educated than average, self-selected, population, whose material circumstances and health perceptions may differ substantially from the overall population. Findings related to these countries are therefore not generalisable to the whole population and need to be interpreted with caution. Another potential source of bias is the restriction to participants with internet access. Despite quota sampling and poststratification weighting, online samples may be different from their populations on important unobservable characteristics. Online surveys, however, have been the predominant means of data collection during the COVID-19 pandemic.

Notwithstanding the above considerations, deterioration of perceived health/HRQoL during the first year of the COVID-19 pandemic is consistent with previous online survey–based

studies, of which a limited number collected EQ-5D data in the general adult population [4–7], and compared them to prepandemic convenience samples. In Portugal [5], only a few weeks into the pandemic (29 March to19 April 2020), HRQoL had deteriorated by 3% in the interviewed adult population quarantined at home ($n = 904$); in Morocco [4], a couple of months into the first lockdown (2 to 30 May 2020), perceived HRQoL deteriorated by 5.5% for the interviewed sample ($n = 537$). In the US [6], in mid-2020, changes of perceived overall health varied with respondents' age ($n = 2,746$), with the largest negative impact experienced by younger adults. Specifically, the perceived health of participants aged 18 to 24 deteriorated by 10% compared with a pre-COVID-19 online survey ($n = 2,028$), and by 18% compared to a face-to-face pre-COVID-19 survey ($n = 1,134$). The first 8 weeks of lockdown worsened perceived overall health of Belgian ($n = 2,099$) and Dutch ($n = 2,058$) adults by 4% and 1%, respectively, compared with prepandemic norms [7]. The magnitude of these perceived changes in overall health, as measured by the EQ-5D, varied across studies in the first year of the pandemic, becoming generally larger as the pandemic unfolded, with the direction of changes fairly consistent and in line with our findings. Only in China did perceived overall health not significantly change in our sample. A similar result was reported in another study using the EQ-5D [10] and attributed to different cultural perceptions of health, overall health status, age, and gender structures, as well as pandemic stages.

Anxiety/depression was the EQ-5D domain that deteriorated most during the COVID-19 pandemic. The mental health impact is consistent across the whole COVID-19 empirical literature and was consistently reported in studies on HRQoL using the EQ-5D, both in the general population [4–7,10,11,33] and in specific clinical subpopulations [9]. The subgroups for which anxiety/depression deteriorated most were female/other gender, and younger people, which is consistent with other published results [5,6,33]. Importantly, the deterioration of mental health was prevalent across all countries regardless of the level of economic development, while for other EQ-5D health domains (S5 Fig), there was a clear inverse country income-health gradient. The fact that the percentage of males who reported deterioration in EQ-5D domains other than anxiety/depression was greater than the corresponding percentage of females/others is in line with previous evidence, as a multicountry study of EQ-5D population norms [42], when investigating responses to the EQ-5D domains by gender, found that usually gender-related ORs were in favour of men in terms of reported problems, but there were some exception in the domains of mobility, self-care, and usual activities in some countries. In the context of the COVID-19 pandemic, these findings may be attributed to gender differences in the questionnaire interpretation. It may in fact be possible that males were interpreting COVID-19-related restrictions (i.e., lockdowns) as limiting their mobility and usual activities, whereas females/others responded to the questions in terms of the extent to which their physical and mental health conditions impacted their physical ability to be mobile and conduct daily activities. However, in the absence of a follow-up question in the survey tailored to ascertain this potential misinterpretation of the EQ-5D questionnaire and the potential gender-related bias in interpretation, this remains only a speculation.

Perceived deterioration of health was significantly associated with higher educational level, the largest odds found for females/others. Higher educated women are more likely to work, but the prolonged COVID-19 lockdowns have increased the burden of unpaid care, which usually disproportionally falls on women [43]. This, in turn, may have negatively affected their mental health and, therefore, their overall perceived health. Our study—similar to others [5,33]—has shown that mental health deterioration was more prevalent in women. We found significantly increased odds of health deterioration across genders when income losses due to COVID-19 occurred, although no association with employment status. This may suggest that it is the immediate and unexpected loss of income that is associated with perceived worse

health most, especially for those with poor job security [9,11,44,45], the likely driver of the association being a deterioration of mental health. Those with chronic diseases were more likely to report deterioration in HRQoL, which is a consistent finding in COVID-19 studies. A potential explanation of this observed association may include delays in obtaining the necessary healthcare and medicines for specific conditions, together with the anxiety that these difficulties may have generated. Direct and/or indirect (through family/friends) experience of/ exposure to COVID-19 at the individual level significantly increased the likelihood of reporting deterioration in health, in line with similar findings in recent community-based studies in Hong Kong [46] and Germany [33]. Government responsiveness to the COVID-19 crisis at national levels was not significantly associated with worsened health for females/others but was associated with reduced odds of worsened health for males. In previous studies of the association between the COVID-19 pandemic and HRQoL of the general population, macro-level indicators of government responsiveness were not generally included, with only a few studies indirectly exploring the impact of lockdown, either in the general population [4] or in clinical subgroups [9]. It is worth remembering, however, that governments' interventional measures varied with the spread of the virus. The way in which the government action was associated with the population's perception of their health, instead, was more likely influenced both by personal and wider contextual circumstances at the time of the survey, as well as by the comparison of both with what previously experienced. For example, strictness of lockdown had generally eased at the time of the survey with respect to the very beginning of the pandemic, although differences among countries existed, which may have affected observed associations. Pandemic severity, proxied by new COVID-19 cases and deaths, and perceived government effectiveness were associated with reduced odds of health deterioration for the male population. As the severity of the virus spread, so did the containment measures adopted by governments, which may have, therefore, confounded these results. Interestingly, higher levels of government effectiveness were associated with reduced likelihood of reporting deterioration of health for males, but the opposite for females/others. It is possible that any positive relationship between government effectiveness and perceived health may be obscured by perception of effectiveness/trust in government. A recent report has found that in most OECD countries, women have lower trust in the national government than men [47]. While a variety of causal mechanisms may drive this finding, women's lower trust in government may derive from lower economic and educational opportunities or the existence of other structural gender inequalities in society. These may have been exacerbated during the COVID-19 pandemic and may have contributed to trigger mental health's deterioration, which, in turn, impacted on overall perceived health. Individual-level trust in governmental actions to face COVID-19 was also found to improve ED-5D-derived HRQoL and mental health in a German study [33], but results were not stratified by gender. Further research is needed to understand the extent to which the observed associations may be causal, and to better elucidate the potential mechanisms underlying those relationships (e.g., mediators and moderators), which was beyond the aims of this study.

Our findings suggest that, when translated into QALYs, the COVID-19 pandemic-related burden associated with morbidity may be substantial compared with plausible QALY loss due to COVID-19 premature death. The advantage of using QALYs to assess the burden of the pandemic, in addition to simpler metrics like confirmed COVID-19 cases/deaths, is that the latter are unable to capture broader pandemic impacts. Those are not only due to decreased access to healthcare and short/long-term deterioration in mental health for non-COVID-19 patients, but also the severity and length of morbidity from COVID-19 itself, including long COVID-19 [48]. As the EQ-5D-5L index can be used to derive QALYs, multicountry longitudinal studies have the potential to capture changes in the general population health profile as

the pandemic unfolds, and the vaccine/booster rollout programme expands globally. This information can provide benchmark evidence for countries at different stages of the pandemic to learn from each other, as well as inform how public health measures and economic policies may be best targeted in the event of other future health shocks.

## Supporting information

**S1 Table. STROBE Statement—Checklist of items that should be included in reports of cross-sectional.**
(DOCX)

**S2 Table. Sociodemographic and clinical characteristics, overall and by country.**
(DOCX)

**S3 Table. Indicators of government responsiveness and perceived effectiveness by country, mean (95% confidence interval).**
(DOCX)

**S4 Table. Paretian Classification of Health Change overall and by continent, mean (95% confidence interval).**
(DOCX)

**S5 Table. Paretian Classification of Health Change by country, mean (95% confidence interval).**
(DOCX)

**S6 Table. Association between worsened health and sociodemographic and clinical characteristics of participants.**
(DOCX)

**S7 Table. Association between worsened health and experiences of/exposure to COVID-19.**
(DOCX)

**S8 Table. Association between worsened health and indicators of responsiveness to and severity of COVID-19.**
(DOCX)

**S9 Table. Association between worsened health and perceived government effectiveness.**
(DOCX)

**S10 Table. Respondents self-reported health on EQ-5D-5L before the COVID-19 pandemic and at time of survey—Overall sample.**
(DOCX)

**S11 Table. Mean difference in EQ-5D-5L index (utility) pre-COVID-19 and at time of survey, UK value set—Male only.**
(DOCX)

**S12 Table. Mean difference in EQ-5D-5L index (utility) pre-COVID-19 and at time of survey, UK value set—Female and Other only.**
(DOCX)

**S13 Table. Mean difference in EQ-5D-5L index (utility) pre-COVID-19 and at time of survey, US value set—Overall sample.**
(DOCX)

**S14 Table. Mean difference in EQ-5D-5L index (utility) pre-COVID-19 and at time of survey, US value set—Male only.**
(DOCX)

**S15 Table. Mean difference in EQ-5D-5L index (utility) pre-COVID-19 and at time of survey, US value set—Female and Other only.**
(DOCX)

**S16 Table. Mean difference in EQ-5D-5L index (utility) pre-COVID-19 and at time of survey, India value set—Overall sample.**
(DOCX)

**S17 Table. Mean difference in EQ-5D-5L index (utility) pre-COVID-19 and at time of survey, India value set—Male only.**
(DOCX)

**S18 Table. Mean difference in EQ-5D-5L index (utility) pre-COVID-19 and at time of survey, India value set—Female and Other only.**
(DOCX)

**S19 Table. National mean difference in EQ-5D-5D-5L index (utility) by age, UK tariff.**
(DOCX)

**S1 Fig. Flow chart of study population.**
(DOCX)

**S2 Fig. Paretian Classification of Health Change (PCHC) by World Bank income classification group (ICG).**
(DOCX)

**S3 Fig. Paretian Classification of Health Change (PCHC) worsened by EQ-5D-5L dimension.**
(DOCX)

**S4 Fig. Paretian Classification of Health Change (PCHC) worsened by EQ-5D-5L dimension and number of long-term health conditions.**
(DOCX)

**S5 Fig. Paretian Classification of Health Change (PCHC) worsened by EQ-5D-5L dimension and ICG.**
(DOCX)

**S6 Fig. Association between worsened health and sociodemographic and clinical characteristics of participants.**
(DOCX)

**S7 Fig. Association between worsened health and experiences of/exposure to COVID-19.**
(DOCX)

**S8 Fig. Association between worsened health and indicators of responsiveness to and severity of COVID-19.**
(DOCX)

**S9 Fig. Association between worsened health and perceived government effectiveness.**
(DOCX)

**S10 Fig. Worsened anxiety/depression by age and gender for each continent.**
(DOCX)

**S11 Fig. Worsened health by EQ-5D domains but anxiety/depression—Stratified for by age and gender.**
(DOCX)

**S12 Fig. Respondents' self-reported health on EQ-5D-5L before the COVID-19 pandemic and at time of survey—Overall sample.**
(DOCX)

## Author Contributions

**Conceptualization:** Mara Violato, Philip M. Clarke.

**Data curation:** Raymond Duch, Matias Fuentes Becerra.

**Formal analysis:** Mara Violato, Jack Pollard.

**Funding acquisition:** Mara Violato, Philip M. Clarke.

**Investigation:** Mara Violato, Jack Pollard, Andrew Lloyd, Laurence S. J. Roope, Raymond Duch, Matias Fuentes Becerra, Philip M. Clarke.

**Methodology:** Mara Violato, Philip M. Clarke.

**Project administration:** Mara Violato.

**Software:** Raymond Duch, Matias Fuentes Becerra.

**Supervision:** Mara Violato.

**Visualization:** Jack Pollard.

**Writing – original draft:** Mara Violato.

**Writing – review & editing:** Mara Violato, Jack Pollard, Andrew Lloyd, Laurence S. J. Roope, Raymond Duch, Matias Fuentes Becerra, Philip M. Clarke.

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
