## [Editor Report · Decision Letter 0]

17 Jun 2022

Dear Dr Violato, 

Thank you for submitting your manuscript entitled "The impact of the COVID-19 pandemic on health-related quality of life: a cross-sectional survey of 13 high and low-middle income countries" for consideration by PLOS Medicine.

Your manuscript has now been evaluated by the PLOS Medicine editorial staff as well as by an academic editor with relevant expertise and I am writing to let you know that we would like to send your submission out for external peer review.

Given the content of your manuscript, the editors would like to propose that it be considered for potential inclusion in the Special Issue (subject to consideration by our panel of guest editors). Your manuscript can still be considered for publication in PLOS Medicine even if you prefer it not to be considered for the Special Issue specifically. Please indicate in your cover letter whether or not you would like the manuscript to be considered for the Special Issue.

Please re-submit your manuscript within two working days, i.e. by Jun 21 2022 11:59PM.

Kind regards,

Callam Davidson

Associate Editor

PLOS Medicine

---

## [Decision Letter · Decision Letter 1]

5 Sep 2022

Dear Dr. Violato,

Thank you very much for submitting your manuscript "The impact of the COVID-19 pandemic on health-related quality of life: a cross-sectional survey of 13 high and low-middle income countries" (PMEDICINE-D-22-01975R1) for consideration at PLOS Medicine as part of our upcoming Special Issue. 

Your paper was evaluated by an associate editor and discussed among all the editors here. It was also discussed with an academic editor with relevant expertise, and sent to independent reviewers, including a statistical reviewer. The reviews are appended at the bottom of this email and any accompanying reviewer attachments can be seen via the link below:

[LINK]

In light of these reviews, I am afraid that we will not be able to accept the manuscript for publication in the journal in its current form, but we would like to consider a revised version that addresses the reviewers' and editors' comments. Obviously we cannot make any decision about publication until we have seen the revised manuscript and your response, and we plan to seek re-review by one or more of the reviewers. 

We hope to receive your revised manuscript by Sep 26 2022 11:59PM. Please email us (plosmedicine@plos.org) if you have any questions or concerns.

We look forward to receiving your revised manuscript. 

Sincerely,

Callam Davidson, 

PLOS Medicine

plosmedicine.org

Please include continuous line numbering throughout your manuscript to facilitate review.

Please update your title to ‘The COVID-19 pandemic and health-related quality of life across 13 high and low-middle income countries: a cross-sectional analysis’, or similar.

PLOS Medicine requires that the de-identified data underlying the specific results in a published article be made available, without restrictions on access, in a public repository or as Supporting Information at the time of article publication, provided it is legal and ethical to do so. Please see the policy at http://journals.plos.org/plosmedicine/s/data-availability and FAQs at http://journals.plos.org/plosmedicine/s/data-availability#loc-faqs-for-data-policy 

Your study is observational and therefore causality cannot be inferred. Please remove language that implies causality, such as ‘impact’ or ‘affect’. Refer to associations instead.

Abstract Background: Provide the context of why the study is important. The final sentence should clearly state the study question.

Abstract Methods and Findings:

* Please ensure that all numbers presented in the abstract are present and identical to numbers presented in the main manuscript text.

* Please include the study design (cross-sectional) and population and setting.

Abstract Conclusions:

* Please address the study implications without overreaching what can be concluded from the data; the phrase "In this study, we observed ..." may be useful.

Please place citations within square brackets and before punctuation. 

Please specify whether informed consent was written or oral.

Please ensure that the study is reported according to the STROBE guideline, and include the completed STROBE checklist as Supporting Information. Please add the following statement, or similar, to the Methods: "This study is reported as per the Strengthening the Reporting of Observational Studies in Epidemiology (STROBE) guideline (S1 Checklist)."

Please provide the rationale behind grouping gender as ‘Male’ and ‘Female & Other’.

Please avoid overstating your findings in the concluding paragraph - temper statement such as ‘Our findings show’ by referring to suggestions, to acknowledge the limitations inherent to the study design.

The ‘Ethics Approval’, ‘Declarations of interests’ and funding information from the ‘Acknowledgements’ section can all be removed, as the relevant information is presented either in the Submission Form or the Methods. 

Ensure date of citation is presented for internet sources in the references. See https://journals.plos.org/plosmedicine/s/submission-guidelines#loc-references for more information. 

Figure S12 – overlapping text makes some of the data labels difficult to read. 

Comments from the reviewers:

Reviewer #1: Thanks for the opportunity to review your manuscript. My role is as a statistical reviewer, so my review concentrates on the study design, data, and analysis that are presented. I have put general questions first, followed by queries relevant to a specific section of the manuscript (with a page/paragraph reference).

This study examines how perceived health-related quality of life changed over the first year of the COVID-19 pandemic, and examines how this was associated with individual and national level characteristics. Participants were recruited online (most countries) and from household sampling of urban communities, with quota sampling used. For some countries, post-stratification weights were created with raking, based on region (within country), age, sex and education. Based on the info from the previous PNAS publication the relative size of the weights would appear to be country specific (e.g. more correction needed for relative under sampling of young age-groups in the USA compared to the other raking variables). CANDOUR is very interesting - and in this manuscript I particularly like the approach of considering 'health' to be more than just incidence of (one) disease and mortality, and the focus on LMICs. 

It looks as though the CANDOUR project and several of the publications have a pre-registered analysis plan available from the project website but I couldn't see one specific for this manuscript. Is one available to be included in the review process?

It might be helpful to be specific about the period over which the perceived change in QoL occurred - the data was collected in late 2020, there might be a standard way to describe this period of time ('early COVID pandemic period'?) or maybe qualify the change was in the first year. 

If post-stratification weights are applied to even part of the data, then the raw numerator for an indicator won't match the proportion reporting a particular indicator. I would just report the %, and preferably with a 95% CI (with your sample size the CI will be precise, nothing wrong with 'flaunting' that). 

Is there any material available which describes the sampling process of the two countries where in-person surveys were completed? Was the sampling frame for these countries truly national or was it based in cities/areas of convenience? I also think that the labels for these two countries might need to be specific about the population, my own experience with survey research in India was that there are non-trivial difference differences between urban and village populations, and also between different areas of the country. 

One limitation of the study was that the change in HR-QOL is based off recall of participants rather than longitudinal measurement. The manuscript is appropriately worded (use of 'perceived deterioration'), but I think the limitations of this measurement should be put into the discussion section so that it's clear.

P2, Paragraph 2. In the methods it describes in-person sampling for two countries - but the description here only mentions the online surveys.

P5, Paragraph 1. What criteria did you use to decide that a sample from a country was imbalanced? 

P7, Paragraph 1. What was the rationale for a) multivariable models, and b) performing variable selection for the multivariable models? I can follow Models 1 and 2, but I was unsure about why the more complex multivariable model and model selection procedure was required. If the aim was to reduce confounding between the covariates, then a criteria of p<0.05 is probably too stringent as it's possible for important levels of confounding to occur where a covariate-outcome relationship has a p-value >0.05. Was p>0.05 also the criteria for the backward selection procedure? For validation of the stepwise procedures, I would suggest a larger p-value and also bootstrapping the stepwise procedure ('swboot' in Stata does this). Was the stepwise proceure needed as there were too many parameters for the sample size? Based on the sample size (n>1000 for each country) is it possible to just include multivariable models with covariates of interest, or were there issues with collinearity or inflated SE?

P8, Paragraph 1. Was missing-as-indicator used for analyses where participants had missing individual-level data? Missing-as-indicator can work for missing-at-random data when there is no correlation between missingness and the values of other covariates (e.g. the treatment effect adjusted for covariates in an RCT). Was this the case for the sample in this study? 

P8, Paragraph 2. I'd agree that the level of missingness in EQ5D is low enough that a complete-case analysis here is ok. 

Reviewer #2: We appreciate the opportunity to review the study entitled "The impact of the COVID-19 pandemic on health-related quality of life: a crosssectional survey of 13 high and low-middle income countries". We believe it addresses a relevant and pertinent issue. The objective of analysing the impact of the COVID-19 pandemic on HRQoL in different countries is fully justified.

The paper provides evidence on the impact of the pandemic on perceptions of health status and highlights the impact on mental health as a key element of this impact. These results are aligned with previously available evidence (Poudel et al Plos One 2021; Nguyen et al J Clin Med 2020,…), but the authors also adjust the effect studied for important personal and contextual factors, and offers an overview of different countries.

However, the study is not without limitations which need to be discussed. 

Our main concerns are related to the so-called "recall bias" and to the utilization of the the measurement tool.

The "recall bias" appears to be significant and it depends on the event being recalled, time since the event, and patient clinical and demographic characteristics. HRQoL seems to be particularly sensitive to this bias. (Schmier et al Expert Rev. Pharmacoeconomics Outcomes Res. 2004). Patients tend to remember their baseline HRQoL as being better than it actually was (Litwin et al, J Clin Oncol 1999). Some of the references used by the authors have measured recall bias in short periods of time, two weeks (Lawson A et al, BMC Musculoskelet Disord 2020). Other studies, such as the one mentioned in the article by Haagsma J, et al (Health and Quality of Life Outcomes. 2019) found no differences over longer time periods, i.e. they did not report a significant recall bias at the group level, but this was due to random error, which did not guarantee the reliability of the data. 

In this study, we consider this bias to be an important limitation, as the questions referred to a period of time that could be almost a year, and in the midst of a situation, the pandemic, that was having a profound social impact. "Recall bias" has been found so crucial in the measurement of HRQoL that adjustment measures have been proposed to "correct" the results, particularly using the EQ-5D (McPhail S et al, Health Qual Life Outcomes 2010). 

Furthermore, the fact that the population norm expresses a better HRQoL status in a country than the selected sample from that country leads us to think about how good the representativeness of the sample was, but does not ensure that the bias does not overestimate the loss of HRQoL. Aditionally, if this situation is a constant in most of the countries studied, it would be worth noting (population norms for the EQ-5D exist in several of the countries studied, Australia, Colombia, China, UK, Canada, Spain...).

The second question is related to the measurement tool. The EQ-5D-5L is a valid and reliable instrument to measure HRQoL. The authors are experts in the use of this tool, but they do not point out in the manuscript that the EQ-5D assesses the current perceived health status. In the context of acute illness, HRQoL may deteriorate significantly, but only temporarily. During the weeks in which the study was conducted, a new peak incidence of infection was occurring worldwide (more than 4 million new cases in Europe alone, COVID-19 Weekly Epidemiological Update), which may have had some influence on the results expressed. But beyond this circumstance, the perception of HRQL may have been more similar to the baseline, further away from the incidence peaks, which has had an extraordinary social impact.

We believe that the impact of the pandemic on HRQoL would be more realistically measured in the nadir periods of the spread of the virus, and especially when the social conditions experienced (and reported by the media) were returning to normal.

This does not detract from the value of the manuscript, but its conclusions must be assessed in the context in which the study was conducted.

Other issues that should be reviewed are listed below:

- Please, explain how were estimated lost QALYs due to deaths. Are they taken from reference 22?

- We are confident that the authors could handle more appropriate methods of analysis when contextual variables are included in addition to individual variables. It would be interesting to know how much of the variability of the response is determined by natural groupings (countries). On the other hand, it would be useful to show some data about the goodness of fit of the models and why they were chosen.

- Wherever possible, country-specific value sets should be used to estimate country-specific utilities.

- Surprising results need to be discussed further. For example, why do the authors believe that higher levels government effectiveness were associated with reduced likelihood of reporting deterioration of health for males, but the opposite for females/others? Why might government effectiveness be related to worsening health? 

- Government responses at national level and government performance indices deserves a more detailed explanation (it seems appropriate to include it in the supplementary material). As pointed out by the authors, some indices, for example those related to government intervention (Government Response Index, Containment and Health Index ...) varied with the spread of the virus while the impact of government action and its relation to the population's perception is conditioned by the overall situation. So, the "lockdown" eased during the months in which the study was conducted, but it was more severe (and different between countries) in the previous months. This situation should be highlighted in the discussion.

- Loss of income, along with previous illnesses, were strongly associated with worsening health status . Could it not be discussed how much of the worsening health status may be due to the economic impact of the pandemic (perhaps mediated by worsening mental health)?

- The representativeness of the selected populations is rather questionable. It is not plausible that there are more highly educated people in Africa than in Europe or in North America. While the authors acknowledge this possible bias, they should discuss at length how it might affect the results.

In conclusion, we believe that this is a valuable study, the results of which may be of interest. However, it would benefit from some improvements in the method (proposed above) and a broader discussion of some of the associations found and the limitations intrinsic to the study design.

Signed: J. Martín-Fernández PhD, MD

Reviewer #3: This is an interesting and informative paper. I have following comments on the paper: 

1. My first suggestion is- please cite all the references within sentences. For example, you have cited references either outside full-stops (.) or outside commas (,) which do not look right.

2. In the Methods section, under 'Procedures', you have mentioned- "QALYs lost at population levels were estimated from the UK-valued EQ-5D-5L and external data on population sizes." Could you please give justifications for this as the value sets may differ greatly by study country? This may affect validity of your analysis results. 

3. In the Methods section, under 'Statistical Analyses', in the last paragraph you have mentioned- "The resulting country level QALY loss due to COVID-19 pandemic-related morbidity was then expressed as a ratio of country-level QALY loss due to COVID-19 premature mortality, estimated by multiplying confirmed country-level COVID-19 deaths by current estimates of QALYs lost per death.(22-24)". Method explained in this sentence is not very clear to readers. Please expand how did you calculate QALYs lost per death for every country, to make it clear to readers? 

4. In addition to above comment, in Table 3, you have used same lower and upper QALYs lost per death values for every country. How do you justify this as we know this may differ in every country?

5. In the Results section (page 10), you have reported - "For all other health domains, younger groups remained most impacted, but deterioration in health was, on average, greater for males (Figure S11)." Could you please explain the reasons why deterioration in health was greater for males than females?

6. In the Results section, under "HRQoL changes" (page 11), you have mentioned- "Only China showed non-statistically significant changes". Why? Do you know any reasons why China showed non-statistically significant changes?

7. In Table 1, could you please explain why female and other gender categorised in one group? Also, what types of gender come under 'other', any examples? Please define 'other' gender in the body of the paper too so that readers are clear about it.

8. In Table 1, average age of respondents from Africa reported as 29.11 yrs but other continent over 42 yrs. I doubt the sample taken from Africa may be representative of the study population (may be due to that reason, income loss from Africa is very high (88.73%) compared to other countries, as shown in Table 1). What is your opinion about the representativeness of sample from Africa? How do you convince the sample from Africa is representative of the study population?

9. Again in Table 1, in co-morbidities categories, one category is- 'Allergies', which is too vague to understand by readers. Can you please give some examples in a bracket what are covered under 'Allergies' so that readers are clearer about this? 

Thank you.

[LINK]

---

## [Decision Letter · Decision Letter 2]

8 Nov 2022

Dear Dr. Violato,

Thank you very much for re-submitting your manuscript "The COVID-19 pandemic and health-related quality of life across 13 high and low-middle income countries: a cross-sectional analysis" (PMEDICINE-D-22-01975R2) for review by PLOS Medicine.

I have discussed the paper with my colleagues and the academic editor and it was also seen again by three reviewers. I am pleased to say that provided the remaining editorial and production issues are dealt with we are planning to accept the paper for publication in the journal.

[LINK]

We look forward to receiving the revised manuscript by Nov 15 2022 11:59PM.   

Sincerely,

Callam Davidson, 

Associate Editor 

PLOS Medicine

plosmedicine.org

Requests from Editors:

Thank you for including an Author Summary - this should immediately follow the Abstract in your revised manuscript.

Please combine bullets one and two in your Author Summary into a single bullet (“The health burden of the COVID-19 pandemic has been closely tracked in terms of confirmed cases and deaths due to the virus, but data on the relationship between the COVID-19 pandemic and health-related quality of life of the general population globally is still quite limited”) to ensure the Author Summary matches our formatting guidelines.

Please change ‘due to’ to ‘associated with’ in bullet four of you current Author Summary to reflect the observational nature of the study.

Thank you for your explanation as to why those identifying as neither male nor female were grouped with females for the analysis. I would suggest updating the additional sentence at line 184 to ‘Because of small numbers, the respondents who reported their gender as ‘Other’ were arbitrarily combined with those who identified as ‘Female’ for the purposes of the analysis’. 

Related to the above, I’d recommend adding footnotes to Table 1 and Figure 2 briefly explaining the grouping of ‘Female & Other’. 

Line 96 is missing a comma between Canada and Norway.

Comments from Reviewers:

Reviewer #1: Thanks for the revised manuscript and responses to my review. Apologies for missing for some of the details in my first revision, the manuscript explains or link to explanations about the study sample and pre-registered protocol. 

One of the other reviewers also noted the interpretation of change in HRQoL based on recall. I think that the limitations of this are fully explained in the manuscript now. While having a time machine to gather this information pre-COVID would be nice, recall is the next best thing and the authors have been clear about what they have collected and the limitations in interpretation. 

The part looking at association between worsened health and individual/country characteristic and experience of COVID looks good to me now, I found the new version easy to follow and understand without the model that had the additional backward selection. With a relatively large sample size using p<0.10 for screening should be ok here. 

Happy to recommend that the paper is accepted.

Reviewer #2: Thank you for allowing us to evaluate the revised version of this interesting manuscript.

We consider that almost all the issues raised have been addressed in a fully satisfactory way. We remain of the opinion that other methods of analysis could provide valuable additional information on the variability of response across countries, but the authors have chosen otherwise.

We are also aware of the limitations of the study in translating its conclusions into decision-making or assessing health situations such as those studied. But the authors have used the available tools in an absolutely accurate and honest way. While the manuscript was certainly valuable in its original presentation, we believe that the authors have improved many important aspects in this review. They have explained, throughout the different sections of the manuscript, all these limitations and their possible implications, so we only have to thank them for their work.

Reviewer #3: The authors have addressed all of my comments. Therefore, I am happy the editorial team take further steps for publication. Thank you.

[LINK]

---

## [Editor Report · Decision Letter 3]

21 Nov 2022

Dear Dr Violato, 

On behalf of my colleagues and the Academic Editor, Dr Lola Kola, I am pleased to inform you that we have agreed to publish your manuscript "The COVID-19 pandemic and health-related quality of life across 13 high and low-middle income countries: a cross-sectional analysis" (PMEDICINE-D-22-01975R3) in PLOS Medicine.

PRESS

PLOS frequently collaborates with press offices. If your institution or institutions have a press office, please notify them about your upcoming paper at this point, to enable them to help maximise its impact. If the press office is planning to promote your findings, PLOS would be grateful if they could coordinate with medicinepress@plos.org. As this manuscript is to be published as part of the upcoming Special Issue on the pandemic and global mental health, it will be opted out of the early version process. If you would like to discuss this further or if you have any further questions or concerns, please reach out directly (cdavidson@plos.org).

Sincerely, 

Callam Davidson 

Associate Editor 

PLOS Medicine